# Influence of N-Acetyl-L-Cysteine on the Pharmacokinetics and Antibacterial Activity of Marbofloxacin in Chickens

**DOI:** 10.3390/antibiotics14040393

**Published:** 2025-04-10

**Authors:** Albena Roydeva, Nikolina Rusenova, Aneliya Milanova

**Affiliations:** 1Department of Pharmacology, Animal Physiology, Biochemistry and Chemistry, Faculty of Veterinary Medicine, Trakia University, 6000 Stara Zagora, Bulgaria; albena.roydeva@trakia-uni.bg; 2Department of Veterinary Microbiology, Infectious and Parasitic Diseases, Faculty of Veterinary Medicine, Trakia University, 6000 Stara Zagora, Bulgaria; nikolina.rusenova@trakia-uni.bg

**Keywords:** chickens, marbofloxacin, N-acetyl-L-cysteine, drug–drug interactions

## Abstract

**Background/Objectives**: Marbofloxacin, a second-generation fluoroquinolone, is used to control economically significant poultry diseases caused by pathogenic bacteria such as *Staphylococcus aureus* and *Escherichia coli*. Although synergistic antimicrobial activity between fluoroquinolones and N-acetyl-L-cysteine (NAC) has been observed in vitro, data on their pharmacokinetic interactions in vivo remain limited. This study aimed to evaluate the effect of NAC on the oral pharmacokinetics of marbofloxacin in broiler chickens and its antibacterial activity against *E. coli* ATCC 25922 and *S. aureus* ATCC 25923, assessing the potential benefits of their combined administration. **Methods**: The pharmacokinetics of marbofloxacin was evaluated in broilers (5 mg/kg dose) after a single intravenous (n = 12) or single oral (n = 12) administration into the crop. The protocol for the co-administration of marbofloxacin and NAC (400 mg/kg via feed) was as follows: on the first day, the poultry (n = 12) received a single oral dose of marbofloxacin via the crop and over the next four days the fluoroquinolone drug was administered via their drinking water. The plasma levels of the drugs were determined using LC-MS/MS analyses, and minimum inhibitory concentrations were determined using the microbroth dilution method. **Results**: NAC significantly reduced the bioavailability of marbofloxacin after a single oral administration into the crop and decreased the elimination rate constant following the administration of both drugs. At a concentration of 20 μg/mL, NAC led to a 3.8-fold reduction in the MIC of marbofloxacin against *E. coli* ATCC 25922 and a 2-fold decrease at concentrations between 1 μg/mL and 6 μg/mL, while no change was observed in marbofloxacin’s effect on *S. aureus* ATCC 25923. **Conclusions:** Oral co-administration of NAC and marbofloxacin reduced the fluoroquinolone’s bioavailability by two-fold while enhancing its antibacterial activity against *E. coli* ATCC 25922.

## 1. Introduction

Marbofloxacin is a second-generation fluoroquinolone with a broad antibacterial activity against many Gram-negative aerobic bacteria and some Gram-positive bacteria and is registered for use in veterinary medicine [1]. It exhibits concentration-dependent bactericidal activity with a significant post-antibiotic effect by inhibiting bacterial DNA topoisomerases II and IV [1,2]. A high oral bioavailability, large volume of distribution, and long half-life combined with short withdrawal periods are among the pharmacokinetic characteristics of oral marbofloxacin that make it an effective therapeutic drug against susceptible bacterial infections in bird species such as chickens [2,3,4,5], turkeys [6], Japanese quails, common pheasants [7], and geese [8,9]. It is considered a critically important veterinary antimicrobial agent by the World Organisation for Animal Health in the treatment of respiratory and enteric diseases in poultry, ruminants, equines, rabbits, and swine [10]. Moreover, marbofloxacin is generally considered safe for use in exotic birds, as no or only mild reversible adverse effects have been reported [11].

Recent studies have reported an increased risk of development of resistance to fluoroquinolones, including marbofloxacin, due to their widespread use against pathogens such as *Staphylococcus aureus* (*S. aureus*) and *Escherichia coli* (*E. coli*), which cause economic losses in poultry husbandry [12,13,14]. As a result, efforts have been made to promote the rational use of antibiotics and reduce the risk of resistance selection and spread, as reflected in EU legislation. According to this, marbofloxacin, as a fluoroquinolone, is classified as a Category B antibacterial drug, and its use is limited to cases where no alternative, clinically applicable, and effective Category C or D antibiotics are available [15]. Its use must be based on antimicrobial susceptibility testing when possible [15].

The emergence and spread of antibiotic resistance require additional strategies to limit their impact, alongside reducing the use of antibacterial drugs, to mitigate negative effects on animals and the environment [16]. One such approach is the combined administration of non-antibiotic compounds to overcome bacterial resistance [17,18,19]. N-acetyl-L-cysteine (NAC) has been shown to exhibit antioxidant properties in poultry [20,21,22,23] and holds potential for use in combination with antibiotic treatments to reduce the risk of developing bacterial resistance [24,25,26]. Furthermore, NAC has been reported as a potential modulator of antibiotic activity [27,28,29,30]. Although the in vitro potentiation of antibacterial activity by co-administering NAC with certain antibacterial drugs, including fluoroquinolones, has been documented, it remains unclear whether these effects occur in vivo. Furthermore, no data are available on the potential pharmacokinetic interactions, which may differ from the in vitro findings.

Considering the responsibility associated with the administration of fluoroquinolones in farm animals and the lack of information on possible interactions between marbofloxacin and N-acetyl-L-cysteine, this study was designed to evaluate the effect of their co-administration on the oral pharmacokinetics and antibacterial activity of marbofloxacin in broiler chickens, as well as to assess the potential benefits of combining them.

## 2. Results

The simultaneous administration of NAC and marbofloxacin for five consecutive days did not result in any clinical manifestations of undesirable effects.

### 2.1. Pharmacokinetic Analysis of Marbofloxacin with and Without NAC

The plasma concentrations of marbofloxacin are presented in Figure 1a,b and Appendix A, while the pharmacokinetic parameters are summarized in Table 1. A significantly lower AUC_0–∞_ value and a higher MRT value were observed after a single oral administration of marbofloxacin (second group) compared to the intravenous treatment (first group). Additionally, the results showed a significant increase in the elimination rate constant (*p* < 0.05) and a significant decrease in AUC_0–∞_ following the first oral dose of marbofloxacin administered into the crop via a plastic tube and after pre-treatment with NAC (third group compared to the first and second groups). The values of C_max_, AUC_0–∞_, F, and MAT for the fluoroquinolone drug were significantly decreased with NAC co-administration. Additionally, a significantly lower elimination rate constant and a prolonged elimination half-life were observed after five days of fluoroquinolone administration via drinking water in combination with N-acetyl-L-cysteine compared to all the other treatments. The median value of the fluctuating marbofloxacin concentrations after multiple doses of the drug did not exceed 12.2% (2.87–48.44). The value of the accumulation index was 1.22 (1.01–1.73).

Low plasma protein binding values for marbofloxacin were found for the medium concentrations: 7.99 ± 1.01 and 3.28 ± 1.1 for 1 and 2.5 µg/mL, respectively. Higher values were observed for low and high fluoroquinolone levels: 24.65 ± 0.4% and 10.39 ± 3.01% for 0.1 and 5 µg/mL, respectively.

### 2.2. Antimicrobial Activity of Marbofloxacin with and Without NAC

An important aspect of evaluating drug–drug interactions involving antibacterial compounds is the need to analyze not only the potential changes in pharmacokinetics but also the possible variations in antibacterial efficacy. Table 2 presents the data for the MIC values of marbofloxacin against the Gram-negative *Escherichia coli* ATCC 25922 and Gram-positive *Staphylococcus aureus* ATCC 25923 strains, used alone or in combination with different concentrations of NAC. N-acetyl-L-cysteine reduced the MIC value of marbofloxacin by 3.8-fold when administered at 20 μg/mL. Concentrations between 6 μg/mL and 1 μg/mL resulted in a two-fold decrease in the MIC of the fluoroquinolone. However, the MIC value against Staphylococcus aureus ATCC 25923 remained unchanged after bacterial incubation with the marbofloxacin–NAC combination. The MBC value for *Escherichia coli* ATCC 25922 was equal to the MIC value, and for *Staphylococcus aureus* ATCC 25923, the MBC value was two-fold higher than the MIC (Table 2).

## 3. Discussion

The present study aimed to evaluate the effect of NAC on the pharmacokinetics and antibacterial activity of marbofloxacin after multiple oral doses in broiler chickens and the potential benefits and drawbacks of the combination. No deviations from normal behavior or signs of pain or distress were observed in the chickens subjected to either the single or combined oral administration of marbofloxacin and NAC. No adverse effects have been reported in previous studies on broiler chickens following the administration of marbofloxacin or N-acetyl-L-cysteine alone [11,31,32].

A non-compartmental analysis was used to characterize the pharmacokinetics of marbofloxacin following the single intravenous and single oral administration alone. Previous studies in chickens reported relatively higher Vz values, ranging from 1.3 to 2.5 L/kg [2,5,33], compared to our observed range of 0.662–1.118 L/kg. The Cl_B_ values in our study (0.116–0.204 L/h/kg) are comparable to those reported in similar investigations (0.19 ± 0.02 L/h/kg in [33]). Taken together, the Vz and Cl_B_ data resulted in a t_1/2el_ of 3.48–4.35 h, which is slightly lower than the previously reported range of 4.89–5.55 h [2,5,33]. A similar trend was observed for the MRT values: 3.25–6.12 h in our study and a range of 6.09 to 7.78 h reported in comparable experiments in chickens [2,5,33]. The slight differences can be explained by the different methods of analysis, breeds, and ages of the poultry, and the use of HPLC versus LC-MS/MS.

The pharmacokinetic parameters following a single oral administration of marbofloxacin into the crop in this study are consistent with the values reported in the available literature. The published values of bioavailability (60.22% to 88.0%) are similar to our findings [2,3,34]. The observed high bioavailability of marbofloxacin is attributed to the inherent lipophilicity of fluoroquinolones, which facilitates their absorption. The reported mean C_max_ and T_max_ values range from 2.11 to 2.19 µg/mL and 0.83 to 1.68 h, respectively [2,32,34]. However, our study showed greater variability in these parameters, with C_max_ and T_max_ values ranging from 1.95 to 5.01 µg/mL and 1 to 6 h, respectively. The data showing a low percentage of plasma protein binding, ranging from 3% to 24%, suggest that marbofloxacin concentrations and AUC values are unlikely to be significantly affected by a decrease in the free drug concentration. The reported mean t_1/2el_ values (4.13–4.89 h) are in close agreement with our observations [32,34,35]. Similarly, the MRT values found in previous studies (ranging from 5.37 to 7.48 h) are comparable to our findings [5,34,35]. In summary, the pharmacokinetic parameters obtained in our experiments are consistent with previously published data on marbofloxacin and align with the typical characteristics of fluoroquinolones.

NAC significantly affected the oral pharmacokinetics of marbofloxacin in broiler chickens. In the group of chickens that received marbofloxacin directly into the crop after pre-treatment with NAC, the drug reached significantly lower C_max_ values compared to the group treated with a single dose of marbofloxacin alone (*p* < 0.05). The changes in the C_max_ can be explained by the effect of marbofloxacin in combination with NAC on the absorption rate [36]. The lower MAT values suggest a faster absorption rate in the presence of NAC compared to the administration of marbofloxacin alone. The decreased C_max_ value corresponded to a two-fold reduction in the AUC_0–∞_, leading to a statistically significant decrease in bioavailability. This indicates a lower systemic exposure to the fluoroquinolone when it is co-administered with NAC in broiler chickens. Fluoroquinolone molecules are characterized by the presence of both acidic and basic groups, which can exist in different protonated forms. These forms vary in solubility depending on the environmental conditions [37]. As a weak organic acid with a pKa of 3.24, NAC can influence the pH in the intestinal lumen, potentially altering the ability of marbofloxacin to cross the intestinal barrier and reducing its absorption [38]. A decrease in the C_max_ and AUC of marbofloxacin in broiler chickens has been reported after pre-treatment with lactic acid, which has a pKa of 3.8 [39]. The changes in the pharmacokinetics of marbofloxacin observed after a single oral administration were also seen after multiple treatments with the fluoroquinolone in combination with NAC. The data showed that steady-state plasma levels (C_avg_) were lower than the C_max_ values, revealing decreased exposure to the fluoroquinolone drug. The significantly longer elimination half-life after multiple doses of marbofloxacin can be explained by the administration of the antibacterial drug via drinking water and the free access and consumption of the medicated water by the chickens. The small fluctuations in drug concentrations during the dosage interval and the low accumulation index indicate a relatively stable exposure over time without significant variations or the risk of excessive drug accumulation.

Optimizing the dosing regimens for antibacterial drugs co-administered with other medications requires consideration of not only the pharmacokinetic drug–drug interactions but also of the impact of the combination on the antibacterial activity. The literature data support the enhanced antibacterial activity of fluoroquinolones, such as enrofloxacin, as well as other antibacterial agents, including beta-lactam antibiotics, apramycin, gentamicin, and tigecycline, when combined with NAC [24,25,26]. These changes in the efficacy of antibiotics in combination with NAC highlight the importance of properly selecting the antibacterial drugs to combine with NAC to achieve a synergistic effect. Combination therapy using NAC and fluoroquinolones is one approach to improving the efficacy of fluoroquinolones and reducing the risk of resistance development [25]. In the present study, the co-administration of marbofloxacin and NAC resulted in a reduction in the minimum inhibitory concentration (MIC) of marbofloxacin against *E. coli* ATCC 25922, providing an effect at lower antibiotic concentrations. Our data showed that the MIC of marbofloxacin against *E. coli* ATCC 25922 decreased in the presence of NAC at concentrations of 1–6 μg/mL. According to our previous study on NAC pharmacokinetics in healthy broiler chickens, slightly higher plasma levels were observed 24 h after its oral administration at a dose of 400 mg/kg BW via the feed, with a C_max_ of 5.74 μg/mL (range: 3.44–9.32 μg/mL) [40]. The observed reduction in the MIC of marbofloxacin against Gram-negative *E. coli* ATCC 25922 from 0.0156 to 0.008 μg/mL, along with an average plasma concentration of 0.41 μg/mL, indicates that the NAC-pre-treated broiler chickens maintained plasma marbofloxacin levels that were 50-fold higher than the MIC throughout the dosing interval. A higher reduction in the MIC was achieved when the fluoroquinolone drug was combined with NAC at a concentration of 20 μg/mL. Similar values of C_max_ (34.18 μg/mL; range: 19.14–57.19 μg/mL) can be reached after direct application of NAC into the crop [40]. The low oral bioavailability and rapid elimination of NAC limits the maintenance of these plasma levels [26,28,41,42]. Additionally, the study by Petkova et al. [30] reveals that the MIC of NAC is very high (>4000 µg/mL) for *S. aureus* ATCC 25923, *S. aureus* O74, *E. coli* ATCC 25922, and *Pseudomonas aeruginosa* ATCC 27853 [30]. Taken together, the pharmacokinetic data of NAC and its MIC values suggest that NAC alone cannot achieve effective systemic concentrations. However, the cited study reported no change in the MIC values of doxycycline when combined with NAC, but it did increase the minimum biofilm inhibitory concentration (MBIC) for *E. coli* ATCC 25922 [30]. This combination also resulted in increased MIC values against *S. aureus* ATCC 25923 and *S. aureus O74*, while showing no change in MBIC against the tested Gram-positive strains and *Pseudomonas aeruginosa* ATCC 27853 [30]. Further investigations are needed to determine NAC levels and its potential antimicrobial activity in the gastrointestinal tract after oral administration alone or in combination with other antibacterial drugs. Additionally, studying the antibacterial activity of NAC against more bacterial strains will provide further insight into its potential clinical applications. The data from our study suggest that oral application of marbofloxacin in combination with NAC could be used for the treatment of gastrointestinal *E. coli* infections in poultry. The limitations include the lack of pharmacokinetic data on the NAC combination in sick animals using a Gram-negative bacterial infection model. Future studies are needed to ensure that therapeutic drug concentrations are achieved and maintained under infection conditions, to determine the optimal dosing regimen, and to confirm the safety of the combination. Additionally, in vitro testing of the marbofloxacin–NAC combination on other strains, including field isolates, could provide further insight into its efficacy. A disadvantage of this combination is the low oral bioavailability of NAC, which limits its use in treating systemic *E. coli* infections. Based on the pharmacokinetic interaction data, it can be concluded that the decreased bioavailability may compromise the systemic efficacy of marbofloxacin, particularly against less susceptible pathogenic bacteria.

## 4. Materials and Methods

### 4.1. Drugs and Reagents

Marbofloxacin (Marfloxin 100 mg/mL injectable solution; KRKA, Novo mesto, Slovenia) was diluted with sterile pyrogen-free water to a concentration of 2% for intravenous (i.v.) administration. The same sterile formulation was diluted to 1% and used for oral administration. N-acetyl-L-cysteine (TLC grade ≥ 99%; Sigma-Aldrich, St. Louis, MO, USA) was applied orally to poultry by mixing it with their feed. The LC-MS/MS analysis of the marbofloxacin plasma concentrations was performed using enrofloxacin hydrochloride, as the internal standard, and a marbofloxacin standard (≥98%; Sigma-Aldrich, St. Louis, MO). The mobile phases were prepared using acetonitrile (LC/MS-grade; Honeywell, Fluka™, Germany), formic acid for mass spectrometry (LC/MS purity: ~98%; Honeywell Fluka™, Seelze, Germany), and water for chromatography (LC-MS-grade; LiChrosolv^®^, Merck KGaA, Darmstadt, Germany).

*Staphylococcus aureus* American Type Culture Collection (ATCC) 25923 and *Escherichia coli* ATCC 25922 were obtained from the Bulgarian National Collection for Microorganisms and Cell Cultures (NBIMCC, Sofia, Bulgaria). Microbiological tests were performed using cation-adjusted Mueller–Hinton broth (MHB; HiMedia Laboratories GmbH, Einhausen, Germany).

### 4.2. Animals and Experimental Design

The study was conducted after receiving ethical approval from the Bulgarian Food Safety Agency (License No. 339; 13 December 2022). All animal studies were performed according to the requirements of Bulgarian legislation (Ordinance 20/01.11.2012).

Thirty-six-day-old Ross hybrid broiler chickens (Cornish ♀ × Plymouth Rock ♂) of both sexes were purchased from a commercial hatchery (Zhuliv EOOD, Stara Zagora, Bulgaria) and housed at the Biobase of the Faculty of Veterinary Medicine, Trakia University. The birds were raised under standard management conditions in accordance with the species’ requirements, ensuring that they remained healthy and free from stress and disease. They were fed an antibiotic-free grower and finisher ration according to the requirements for their age, with water provided ad libitum. At four weeks of age, the broiler chickens (n = 36) were randomly divided into three experimental groups, each consisting of twelve birds.

The chickens (n = 12, 1.48 ± 0.11 kg body weight (BW)) from Group I were treated intravenously (i.v.) with the 2% marbofloxacin solution. The poultry received a single dose of 5 mg/kg BW via the left v. subcutanea ulnaris. Blood samples were collected from the right wing vein of the birds (n = 6 chickens at every sampling time) in heparinized tubes at 0.083, 0.25, 0.5, 0.75, 1, 1.5, 2, 3, 6, 9, 12, 14, 24, 30, 36, and 48 h after the administration of marbofloxacin.

The second group (n = 12; 2.0 ± 0.17 kg BW) was treated orally via intraingluvial gavage using a soft probe. The broilers received a single dose (5 mg/kg BW) of the 1% marbofloxacin solution. To eliminate the possibility of food–drug interactions, the birds were deprived of feed 12 h prior to the treatment. Blood samples were collected from either the right or left wing vein at the following time points (n = 6 chickens per sampling time): 0.083, 0.25, 0.5, 0.75, 1, 1.5, 2, 3, 6, 8, 10, 12, 24, 30, 36, and 48 h after drug administration.

The first dose of marbofloxacin (5 mg/kg BW) was administered to the chickens in Group III (n = 12, 1.80 ± 0.16 kg BW) via a soft probe into the crop after two days of NAC pre-treatment at a dose of 400 mg/kg BW, which was mixed with the feed (Figure 1). Over the next four days, the broilers in Group III received marbofloxacin at a dose of 5 mg/kg BW/day through their drinking water, while NAC was administered orally at a dose of 400 mg/kg BW via their feed. NAC administration continued for two days after the final dose of marbofloxacin. Feed and water consumption were measured daily, and the doses of NAC and marbofloxacin were adjusted accordingly. Based on our previous experiment, the received dose was close to the target dose: 405 ± 29 mg/kg body weight [40]. The broiler chickens were fasted for 12 h prior to the first oral administration of marbofloxacin. The doses were calculated based on the average bird weight and water consumption measured the previous day. The sampling times were as follows: 0.083, 0.25, 0.5, 0.75, 1, 1.5, 2, 3, 6, 8, 10, 12, 24, 30, 36, 48, 96, 120, 122, 124, 126, 128, 132, 144, 150, 156, and 168 h after the start of the marbofloxacin treatment.

Blood samples were collected from six chickens in each experimental group at each designated time point. Approximately 0.8 mL of blood was drawn from each chicken per collection time point, centrifuged at 1500× *g* for 10 min, and the plasma was collected and stored at −80 °C until the analysis.

### 4.3. Determination of Marbofloxacin Concentrations by LC-MS/MS Analysis

Marbofloxacin was extracted from 300 μL of plasma. The samples were transferred into 2 mL Eppendorf tubes. Then, 10 μL of enrofloxacin hydrochloride (internal standard) at a concentration of 6 μg/mL was added, resulting in a final concentration of 100 ng/mL. To deproteinize the plasma, 290 μL of 0.1% formic acid in acetonitrile was added, and the mixture was vortexed for 1 min. The mixture was then shaken at 200 g/min for 20 min (Lauda™ Varioshake vs. 8 BE shaker with BS1363 UK-Plug; Marlton, NJ, USA). Following this, it was centrifuged at 14 370× *g* for 15 min at 4 °C. The supernatant was filtered through 0.22 μm syringe filters (Agilent Captiva Econo Filter, PTFE membrane; Santa Clara, CA, USA) and transferred into LC-MS/MS vials for analysis.

The chromatographic separation of the compounds was performed using a Zorbax Eclipce Plus (2.1 mm i.d. × 50 mm, 1.8 µm; Agilent Technologies, USA) connected to a precolumn Zorbax SB-C18 (2.1 × 5 mm, 1.8 µm; Agilent Technologies, USA). The liquid chromatography module consisted of a 1260 Infinity II quaternary pump and a 1260 Infinity II Vial Sampler. The temperature of the column was maintained at 40 °C. The autosampler was set at 8 °C. Mobile phase A consisted of 0.1% formic acid in water, while mobile phase B consisted of 0.1% formic acid in methanol. The following gradient mode was applied: 0–1 min: 98% A/2% B; 1–7 min: 98% A/2% B to 60% A/40% B; 7–11 min: 60% A/40% B to 0% A/100% B; 11–13 min: 0% A/100% B; 13–13.1 min: 0% A/100% B to 98% A/2% B; 13.1–17 min: 98% A/2% B; 17–20 min: 98% A/2% B; and 20–24 min: post-run (98% A/2% B). The flow rate was 0.2 mL/min. The injection volume was 5 μL.

The quantification of marbofloxacin was performed using an Agilent 6460c triple-quadrupole mass spectrometer (Agilent Jet Stream (AJS) Technology, Santa Clara, CA, USA). The analysis was performed by applying the ESI positive ion mode (Agilent Jet Stream ESI+). The following conditions were set: drying gas temperature (N2) of 300 °C; 7 L/min flow for the drying gas (N2); 50 psi nebulizer gas (N2); sheath gas temperature (N2) of 350 °C; sheath flow of 10 L/min; capillary voltage of 3000 V; nozzle voltage of 500 V; and dwell time of 200 ms. The qualifying ion for marbofloxacin was 363.2 *m*/*z*, and the quantitative ions were 320.1 *m*/*z* and 72.1 *m*/*z*. For enrofloxacin, the ions were 360 *m*/*z*, 342.1 *m*/*z*, and 316.2 *m*/*z* [43]. The data analysis and quantification of marbofloxacin was performed using MassHunter 10.0 software (Agilent Technologies, Santa Clara, CA, USA). The retention time of marbofloxacin was 9.7 min.

The quantification of marbofloxacin in the plasma samples was performed in reference to a calibration curve, which showed acceptable linearity over a range of eight different concentrations of the standard (5, 10, 50, 100, 250, 500, 750, and 1000 ng/mL), as indicated by a mean correlation coefficient (R^2^) value of 0.9978. The limit of detection (LOD) for the drug was 0.004 μg/mL. The limit of quantification (LOQ) was 0.013 μg/mL. The accuracy ranged from 91.27% to 109.96%. The intra- and inter-day precision values were below 8.29% and 14.35%, respectively.

### 4.4. Protein Binding

Standard solutions of marbofloxacin in chicken plasma with low (0.1 µg/mL), medium (1 µg/mL and 2.5 µg/mL), and high (5.0 µg/mL) concentrations were used for the determination of the amount of protein bound by the fluoroquinolone drug. Ultrafree MC Centrifugal Filters with a hydrophilic PTFE membrane and 0.45 μm pore size (Merck KGaA, Darmstadt, Germany) were used according to the manufacturer’s instructions. Plasma samples (800 μL) with the different marbofloxacin concentrations added were incubated for 1 h at 37 °C. Afterwards, they were centrifuged at 1000× *g* for 10 min and then at 2000× *g* for 20 min. The filtrate (5 μL) from each sample was analyzed using the described LC-MS/MS method. The tests were performed in triplicate. The percentage of protein binding was determined using the following equation:% protein binding = (CTP − CFP/CTP) × 100,(1)
where CTP is the total plasma concentration and CFP is the unbound concentration in the filtrate [44].

### 4.5. Pharmacokinetic Analysis

The pharmacokinetic analysis was performed using Phoenix 8.3 software (Pharsight Certara, St. Louis, MO, USA). A non-compartmental analysis was applied for the computation of the pharmacokinetic parameters based on the determined plasma marbofloxacin concentrations for every chicken (n = 12 per group). The following parameters were calculated: λ, elimination rate constant; t_1/2el_, terminal half-life; T_max_, time at maximum plasma concentration; C_max_, maximum plasma concentration; C_avg_, average plasma concentration; AUC(_0-inf_), area under the curve from zero to infinity; Cl, total body clearance; Vss, volume of distribution at steady state; Vz, area volume of distribution; MRT, mean residence time; MAT, mean absorption time; F, bioavailability; and AUC_0-τ_, partial area from dosing time to dosing time plus dosing interval. The linear-up/log-down method was used for the AUC calculation after a single oral administration of marbofloxacin alone or in combination with NAC. The value of R2 was >0.943, and the extrapolation of the AUC was lower than 20%.

The bioavailability (F) was calculated according to the following equation:F % = (AUCp.o./AUCi.v.) × 100,(2)
where AUCp.o. and AUCi.v. are the area under the curve after oral or intravenous administration, respectively.

### 4.6. Determination of MIC and MBC Values

The bacterial strains were stored at −80 °C prior to use. The strains were grown on tryptic soy agar (TSA; Sigma-Aldrich, St. Louis, MO, USA, Product of India) supplemented with 5% defibrinated sheep blood. Colonies from an overnight colony were directly suspended in Mueller–Hinton broth (MHB; HiMedia, Mumbai, India) until a turbidity comparable to the McFarland turbidity standard of 0.5 (Densilameter II; Erba Lachema, Brno, Czech Republic) was reached. The broth was used at a ratio of 1:100 to dilute the cultures to a concentration of 10^6^ CFU/mL.

The broth microdilution assay was used to determine the minimum inhibitory concentration (MIC) of marbofloxacin for *E. coli* ATCC 25922 and *S. aureus* ATCC 25923. Serial (two-fold) dilutions of marbofloxacin were prepared in Muller–Hinton broth with an initial concentration of 256 μg/mL and then 100 μL (in the trial without NAC) aliquots of the dilutions were added to the wells of 96-well flat bottom plates (Costar; Corning Incorporated, Kennebunk, ME, USA). Then, 100 μL aliquots of *E. coli* ATCC 25922 and S. aureus ATCC 25923 suspensions prepared in Mueller–Hinton broth with an approximate cell density of 1 × 106 CFU/mL were added to each well. The plates were incubated at 37 °C for 20 h and then the optical density (OD) was measured at a wavelength of 620 nm (Synergy LX Multi-Mode Microplate Reader; BioTek, Winooski, VT, USA). The MIC was defined as the lowest drug concentration resulting in an OD value close to that of the blank. The independent experiments were performed in triplicate.

The determination of the MIC value for each bacterial strain was performed in 96-well plates by adding 50 μL of marbofloxacin at an initial concentration of 256 μg/mL to 50 μL of the bacterial suspensions. Serial 2-fold dilutions of marbofloxacin were prepared. Then, 50 μL of NAC was added to each plate containing the serial dilutions of marbofloxacin, resulting in final NAC concentrations of 1, 2, 4, 6, and 20 μg/mL. The plates were incubated at 35 °C for 20 h. The NAC concentrations used in this study were chosen based on a previous study, which demonstrated that an NAC concentration of 34.18 (19.14–57.19) μg/mL was achievable in chicken plasma after a single oral administration at a dose of 400 mg/kg BW (Roydeva et al., 2024 [40]). The absorbance was measured at 620 nm using a plate reader (Synergy LX Multi-Mode Microplate Reader; BioTek, Winooski, VT, USA).

Aliquots (10 μL) from wells at or above the MIC were subcultured on TSA plates to determine the MBC values. The Petri dishes were incubated at 35 °C for 20 h. The minimum bactericidal concentration (MBC) was defined as the lowest concentration of marbofloxacin at which >99.9% of the inoculated organisms were killed. Each experiment was performed in triplicate.

### 4.7. Statistical Analysis

The results from the non-compartmental pharmacokinetic analysis are presented as the geometric mean and range of the minimum and maximum values. The normal distribution of the data was confirmed using the Shapiro–Wilk test. ANOVA, followed by a Bonferroni post hoc test, was applied for statistical analysis of the data. The comparison of the T_max_ values was performed using the Mann–Whitney test due to the absence of a normal distribution. A *p*-value < 0.05 was considered to be significant (Statistica 10.0; Tibco, Palo Alto, CA, USA).

## 5. Conclusions

The results of this study demonstrated that oral administration of marbofloxacin at a dose of 5 mg/kg BW and NAC at 400 mg/kg BW in feed led to a two-fold decrease in the bioavailability of the fluoroquinolone and a reduction in the MIC for *Escherichia coli* ATCC 25922. Based on these findings, the combination of marbofloxacin and NAC could have beneficial effects in treating localized gastrointestinal infections caused by susceptible Gram-negative microorganisms. However, further studies in sick animals are necessary to confirm the efficacy and safety of this combination.

## Data Availability

Data are contained within the article or Appendix A.

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
