# Peer review of "Influence of N-Acetyl-L-Cysteine on the Pharmacokinetics and Antibacterial Activity of Marbofloxacin in Chickens"

_antibiotics, 2025, doi:10.3390/antibiotics14040393_

Round 1
Reviewer 1 Report
Comments and Suggestions for Authors
The present study describes the effects of the co-administration of the fluoroquinolone antibiotic Marbofloxacin with N-acetyl-L-cysteine in chickens. Both the pharmacokinetic properties and antibacterial activity of Marbofloxacin combined with N-acetyl-L-cysteine were investigated. As a result, it was demonstrated that the administration of NAC reduces the bioavailability of Marbofloxacin by about two-fold while enhancing its antimicrobial activity against E. coli in a dose-dependent manner.
Overall, the manuscript is well-organized, well-written, and the results are thoroughly discussed. Additionally, the findings shed new light on the pharmacokinetics and antimicrobial properties of Marbofloxacin. After a careful reading of the manuscript, I did not find any major issues. Therefore, my suggestion is to accept the paper in its current form.
Author Response
Dear Editor-in-Chief,
Thank you very much for the answer related to our manuscript with ID antibiotics-3549017. We tried to answer to all of the Reviewer’s remarks and to revise the manuscript according to the suggestions.
Reviewer 1
The present study describes the effects of the co-administration of the fluoroquinolone antibiotic Marbofloxacin with N-acetyl-L-cysteine in chickens. Both the pharmacokinetic properties and antibacterial activity of Marbofloxacin combined with N-acetyl-L-cysteine were investigated. As a result, it was demonstrated that the administration of NAC reduces the bioavailability of Marbofloxacin by about two-fold while enhancing its antimicrobial activity against E. coli in a dose-dependent manner.
Overall, the manuscript is well-organized, well-written, and the results are thoroughly discussed. Additionally, the findings shed new light on the pharmacokinetics and antimicrobial properties of Marbofloxacin. After a careful reading of the manuscript, I did not find any major issues. Therefore, my suggestion is to accept the paper in its current form.
Answer: Thank you for the evaluation of the manuscript as well done work!
Reviewer 2 Report
Comments and Suggestions for Authors
The manuscript investigates the effect of N-acetyl-L-cysteine (NAC) on the oral pharmacokinetics of marbofloxacin, a second-generation fluoroquinolone, in broiler chickens. In addition, the authors also investigated the antibacterial activity of marbofloxacin using E. coli and S. aureus cultures, when co-administered with NAC.
The paper explores this novel therapeutic combination and provides insights into improving the treatment of infections in poultry while considering the epidemic of antimicrobial resistance.
While other studies have examined antibiotic combinations to enhance efficacy, authors claim that this study adds value by exploring a combination of NAC with marbofloxacin. This finding could help reduce the required dose of marbofloxacin while maintaining effectiveness. Authors claim that this is a novel study, providing references that substantiate this claim would add more value to the manuscript. The study validates the previously published pharmacokinetic parameters for marbofloxacin, but it goes further by introducing the NAC interaction, which can help modify how these drugs should be dosed in practice.
Stretching Figure 1b horizontally will help making the data more legible and clearer to the readers.
The methodology does not explicitly mention how it was ensured that all the feed containing NAC was consumed by the chickens. Please add a sentence or two to describe how this was ensured.
Incorporating a control group that receives NAC alone should be considered for future studies. Please include a more elaborate plan for future under conclusions such as increasing sample size, assessing a wider range of bacteria, clinical efficacy in infected chickens, etc. that would help provide deeper insights into the practical use of NAC and marbofloxacin combination. Compartmental PK and covariates should also be explored.
Author Response
Dear Editor-in-Chief,
Thank you very much for the answer related to our manuscript with ID antibiotics-3549017. We tried to answer to all of the Reviewer’s remarks and to revise the manuscript according to the suggestions.
Reviewer 2
The manuscript investigates the effect of N-acetyl-L-cysteine (NAC) on the oral pharmacokinetics of marbofloxacin, a second-generation fluoroquinolone, in broiler chickens. In addition, the authors also investigated the antibacterial activity of marbofloxacin using E. coli and S. aureus cultures, when co-administered with NAC.
The paper explores this novel therapeutic combination and provides insights into improving the treatment of infections in poultry while considering the epidemic of antimicrobial resistance.
While other studies have examined antibiotic combinations to enhance efficacy, authors claim that this study adds value by exploring a combination of NAC with marbofloxacin. This finding could help reduce the required dose of marbofloxacin while maintaining effectiveness. Authors claim that this is a novel study, providing references that substantiate this claim would add more value to the manuscript. The study validates the previously published pharmacokinetic parameters for marbofloxacin, but it goes further by introducing the NAC interaction, which can help modify how these drugs should be dosed in practice.
Stretching Figure 1b horizontally will help making the data more legible and clearer to the readers.
Answer: The Figure 1b has been revised.
The methodology does not explicitly mention how it was ensured that all the feed containing NAC was consumed by the chickens. Please add a sentence or two to describe how this was ensured.
Answer: The following explanation was included in the text: “Feed and water consumption were measured daily, and the doses of NAC and marbofloxacin were adjusted accordingly. Based on our previous experiment, the received dose was close to the target dose: 405 ± 29 mg/kg body weight.” (Lines 323-326)
Incorporating a control group that receives NAC alone should be considered for future studies. Please include a more elaborate plan for future under conclusions such as increasing sample size, assessing a wider range of bacteria, clinical efficacy in infected chickens, etc. that would help provide deeper insights into the practical use of NAC and marbofloxacin combination. Compartmental PK and covariates should also be explored.
Answer: Thank you very much for this remark! This study is part of a PhD thesis, and an experiment on NAC pharmacokinetics was conducted one month before the presented study. This previous experiment helped, to a certain extent, in better organizing the current study. The results on NAC pharmacokinetics have been published, and the full PhD thesis will provide further insights.
Reviewer 3 Report
Comments and Suggestions for Authors
Thank you for the opportunity to review your manuscript on “Influence of N-Acetyl-L-Cysteine on the Pharmacokinetics and Antibacterial Activity of Marbofloxacin in Chickens."
Your study provides valuable insights regarding the changes in the pharmacokinetic behavior of marbofloxacin in combination with N-acetyl-L-cysteine (NAC). Overall, the manuscript is well organized and covers important aspects of combination therapy. However, I feel there are several areas that require improvements in clarity and structure to increase the overall quality of this manuscript.
Many sentences in the introduction look too long. I suggest breaking these into shorter sentences for more clarity and to improve readability.
In line 53-54, there is an abrupt transition from the rational use of antibiotics to EU legislation. I suggest adding a connecting sentence to improve the flow.
In line 59: “The emergence and spread of antibiotic resistance require additional strategies,” it feels like strategies are required to improve antibiotic resistance. Please rephrase this to inhibit resistance for more clarity.
In section 2.1 “Pharmacokinetic analysis of marbofloxacin with and without NAC,” pharmacokinetic parameters such as AUCâ‚€–∞ were spread across for different groups (1, 2, and 3). I suggest the authors group these results for clear comparative interpretation.
In table 2, the MIC values for marbofloxacin combined with different concentrations of NAC are reported; however, the MIC for NAC alone was not included as a control. Including this NAC alone could help clarify if the observed effects are from the combination or the NAC alone.
In line 223: the term Synergistic Effect, if this statement is to reflect general findings from the literature, make this clearly distinguished from the current study results. If this is to reflect this study, I would suggest to avoid this term as the effect of NAC alone was not tested.
Comments on the Quality of English Language
I would suggest improving the sentence structure for more clarity and readability.
Author Response
Dear Editor-in-Chief,
Thank you very much for the answer related to our manuscript with ID antibiotics-3549017. We tried to answer to all of the Reviewer’s remarks and to revise the manuscript according to the suggestions.
Reviewer 3
Thank you for the opportunity to review your manuscript on “Influence of N-Acetyl-L-Cysteine on the Pharmacokinetics and Antibacterial Activity of Marbofloxacin in Chickens."
Your study provides valuable insights regarding the changes in the pharmacokinetic behavior of marbofloxacin in combination with N-acetyl-L-cysteine (NAC). Overall, the manuscript is well organized and covers important aspects of combination therapy. However, I feel there are several areas that require improvements in clarity and structure to increase the overall quality of this manuscript.
Many sentences in the introduction look too long. I suggest breaking these into shorter sentences for more clarity and to improve readability.
Answer: The manuscript was subjected to English editing after submission and this issue, I hope, is professionally solved.
In line 53-54, there is an abrupt transition from the rational use of antibiotics to EU legislation. I suggest adding a connecting sentence to improve the flow.
Answer: The following sentence replaced the old one: “As a result, efforts have been made to promote the rational use of antibiotics and reduce the risk of resistance selection and spread, as reflected in EU legislation.” (Lines 53-55)
In line 59: “The emergence and spread of antibiotic resistance require additional strategies,” it feels like strategies are required to improve antibiotic resistance. Please rephrase this to inhibit resistance for more clarity.
Answer: The sentence has been modified: “The emergence and spread of antibiotic resistance require additional strategies to limit their impact, alongside reducing the use of antibacterial drugs, to mitigate negative effects on animals and the environment”
In section 2.1 “Pharmacokinetic analysis of marbofloxacin with and without NAC,” pharmacokinetic parameters such as AUCâ‚€–∞ were spread across for different groups (1, 2, and 3). I suggest the authors group these results for clear comparative interpretation.
Answer: A clarification for the groups has been added in the text (Lines 87-91).
In table 2, the MIC values for marbofloxacin combined with different concentrations of NAC are reported; however, the MIC for NAC alone was not included as a control. Including this NAC alone could help clarify if the observed effects are from the combination or the NAC alone.
Answer: In a previous study, published by our group MIC of NAC were determined and the following sentence was added to the discussion “Additionally, the study by Petkova et al. [30] reveals that the MIC of NAC is very high (>4000 µg/mL) for S. aureus ATCC 25923, S. aureus O74, E. coli ATCC 25922, and Pseudomonas aeruginosa ATCC 27853 [30]. Taken together, the pharmacokinetic data of NAC and its MIC values suggest that NAC alone cannot achieve effective systemic concentrations. However, the cited study reported no change in the MIC values of doxycycline when combined with NAC, but it did increase the minimum biofilm inhibitory concentration (MBIC) for E. coli ATCC 25922 [30]. This combination also resulted in increased MIC values against S. aureus ATCC 25923 and S. aureus O74, while showing no change in MBIC against the tested Gram-positive strains and Pseudomonas aeruginosa ATCC 27853 [30]. Further investigations are needed to determine NAC levels and its potential antimicrobial activity in the gastrointestinal tract after oral administration alone or in combination with other antibacterial drugs. Additionally, studying the antibacterial activity of NAC against more bacterial strains will provide further insight into its potential clinical applications.” (Lines 245-259).
In line 223: the term Synergistic Effect, if this statement is to reflect general findings from the literature, make this clearly distinguished from the current study results. If this is to reflect this study, I would suggest to avoid this term as the effect of NAC alone was not tested.
Answer: The discussion was revised by adding our previous and published data about MIC values of NAC: “Additionally, the cited study reveals that the MIC of NAC is very high (>4000 µg/mL) for S. aureus ATCC 25923, S. aureus O74, E. coli ATCC 25922, and Pseudomonas aeruginosa ATCC 27853 [30]. Taken together, the pharmacokinetic data of NAC and its MIC values suggest that NAC alone cannot achieve effective systemic concentrations. Further investigations are needed to determine its levels and potential antimicrobial activity in the gastrointestinal tract after oral administration.”
Once again thank you for the remarks which helped for improvement of the presented investigation. We tried to revise the manuscript according to the remarks and we hope that the revision is at acceptable level.